# CHEF in the Language Kitchen: A Generative Data Augmentation Leveraging Korean Morpheme Ingredients

**Jaehyung Seo[1], Hyeonseok Moon[1], Jaewook Lee[1], Sugyeong Eo[1]**
**Chanjun Park[2] and Heuiseok Lim[1†]**

[1]Department of Computer Science and Engineering, Korea University
[2]Upstage

[1]{seojae777,glee889,jaewook133,djtnrud,limhseok}@korea.ac.kr
[2]chanjun.park@upstage.ai

## Abstract

Korean morphological variations present unique opportunities and challenges in natural language processing (NLP), necessitating an advanced understanding of morpheme-based sentence construction. The complexity of morphological variations allows for diverse sentence forms based on the syntactic-semantic integration of functional morphemes (*i.e.*, affixes) to lexical morphemes (*i.e.*, roots). With this in mind, we propose a method - **CHEF**, replicating the morphological transformations inherent in sentences based on lexical and functional morpheme combinations through generative data augmentation. CHEF operates using a morpheme blender and a label discriminator, thereby enhancing the diversity of Korean sentence forms by capturing the properties of agglutination while maintaining label consistency. We conduct experiments on Korean multiple classification datasets, improving model performance in full- and few-shot settings. Our proposed method boosts performance beyond the preceding data augmentation methods without incurring external data usage. We demonstrate that our approach achieves comparable results yielded by augmentation techniques that use large language models (LLMs).

## 1 Introduction

As an agglutinative language, Korean encompasses a rich array of functional morphemes (Song, 2006; Park et al., 2018). Deep learning-based research in Korean NLP aims to enhance linguistic construction efficiency through morpheme segmentation. Previous studies have explored morphological analysis to improve model performance (Song and Park, 2019; Lee et al., 2020; Kim and Colineau, 2020; Kim et al., 2022). The Korean language presents distinct challenges and opportunities with respect to transformations contingent upon the combinatory patterns between lexical and functional morphemes (Matteson et al., 2018; Seo et al., 2022).

---
† Corresponding author

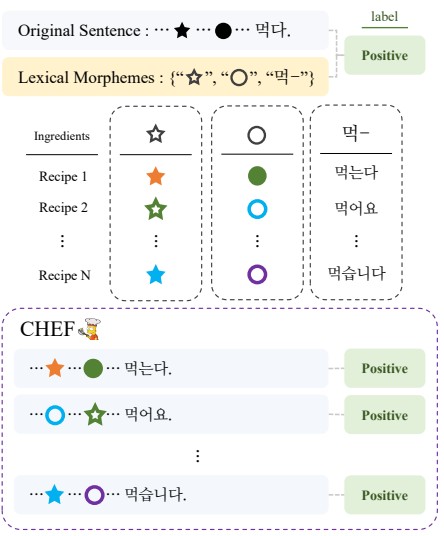

Figure 1: CHEF for data augmentation. The Korean verbs '먹다', '먹는다', '먹어요', and '먹습니다' have equivalent meaning of '**eat**'. The color variations in the shapes indicate morphological changes resulting from combinations with functional morphemes. The 'Recipe N' denotes a morpheme set that can compose various sentences. 'Positive' represents the example of a label attached to the original sentence.

Consequently, the incorporation of morphological alterations following combinatory rules enables a significant diversity of linguistic forms to be generated from given lexical morphemes.

In this sense, Korean datasets are constrained in capturing the intricate rules governing the interactions between lexical and functional morphemes. As a result, the representations derived from these datasets only contain a small fraction of the potential linguistic forms. For instance, Figure 1 demonstrates multiple approaches to represent the concept of 'eat' by employing different ending forms. Additionally, previous data augmentation approaches are rarely designed to generate synthetic data that considers the morphological characteristic of the Korean language, resulting in limited effectiveness in data augmentation. To address these limitations,

it is crucial to develop data augmentation methods that explicitly consider morphological diversity. By incorporating such properties into the augmentation process, we can contribute to the robustness of models and facilitate performance enhancements.

In this paper, we propose CHEF, a data augmentation method designed to construct new synthetic data in alignment with the combinations of lexical and functional morphemes. CHEF is composed of a **morpheme blender**[1] and **label discriminator**. For the initial phase, we prepare Korean lexical morphemes from the training dataset as ingredients. The morpheme blender employs the lexical morphemes and generates the new synthetic sentence by leveraging the knowledge of derivational and inflectional rules in conjunction with functional morphemes obtained from a pre-trained generative model. Morpheme blender can generate diverse sentence expressions by incorporating lexical and functional morphemes, yet it does not guarantee the conservation of label information. To adjust the unintended blending, the label discriminator controls the synthetic data generation process of the morpheme blender. Its primary objective is to prevent substantial shifts in meaning and ensure alignment with the original labels. The teacher forcing of the morpheme blender incorporates global information about synthetic data completed by combinations of lexical and functional morphemes. This is achieved through contrastive learning with the label discriminator. Integrating these two modules, CHEF ensures that synthetic data maintains label consistency while blending new morpheme combinations.

As illustrated in Figure 1, CHEF augmentation enriches text data by attaching various functional morphemes to a given set of lexical morphemes from the original sentence while preserving the label information. The main contributions of our method are summarized as follows:

- Despite difficulties enhancing performance in full-shot settings through data augmentation, CHEF demonstrates effectiveness and robustness through in-depth analysis.

- We observe that employing contrastive learning between a label discriminator and a morpheme blender is a suitable data augmentation approach for maintaining label consistency.

---

[1]Morpheme blender takes morphemes and generates a sentence that contains given morphemes. In this sense, we denote it as a blender.

- CHEF unlocks the morphological diversity in the training data without using any additional external data and approximates or even outperforms the LLM-based data augmentations.

- CHEF exhibits effectiveness even with small amounts of data augmentation.

## 2   Related Work

Data augmentation is widely recognized in deep learning research as a valuable approach to addressing the scarcity of annotated data. It aids in ensuring that the distribution of the training data maintains robustness even in unseen tests. Over time, several methods have been developed to augment textual content, expanding the diversity and quantity of available training data.

Zhang et al. (2015) proposed a method of word substitution using a synonym thesaurus based on rules, which is followed by augmentation methods utilizing a thesaurus in the studies of (Dai and Adel, 2020; Daval-Frerot and Weis, 2020). Wei and Zou (2019) suggested an easy data augmentation (EDA) method using word-level replacement, random insertion, random deletion, and random swap. Karimi et al. (2021) introduced an easier data augmentation (AEDA) technique, improving text classification performance by randomly inserting six pre-defined punctuation marks. Another strategy involves Mixup-based data augmentation (Zhang et al., 2018), combining two sample data to create new training data. Mixup augmentation was initially applied to image data and extended to text-based deep neural networks (Guo, 2020). Subsequently, it has also been applied to textual data augmentation using Transformer-based methods, serving as a technique to mitigate model overfitting and enhance generalization capabilities (Sun et al., 2020; Yoon et al., 2021; Kong et al., 2022).

Sennrich et al. (2016) proposed a back translation method for augmenting data by performing a round-trip translation on data written in the original language, using a neural machine translation system trained on human-annotated data. Data augmentation based on back translation has evolved as one of the crucial techniques since it allows for rewriting the entire sentence, rather than just word-level alterations (Fabbri et al., 2021; Lowell et al., 2021; Park et al., 2021a).

Recently, research on data augmentation using pre-trained language models has been actively pursued (Kumar et al., 2020; Du et al., 2021; Schick

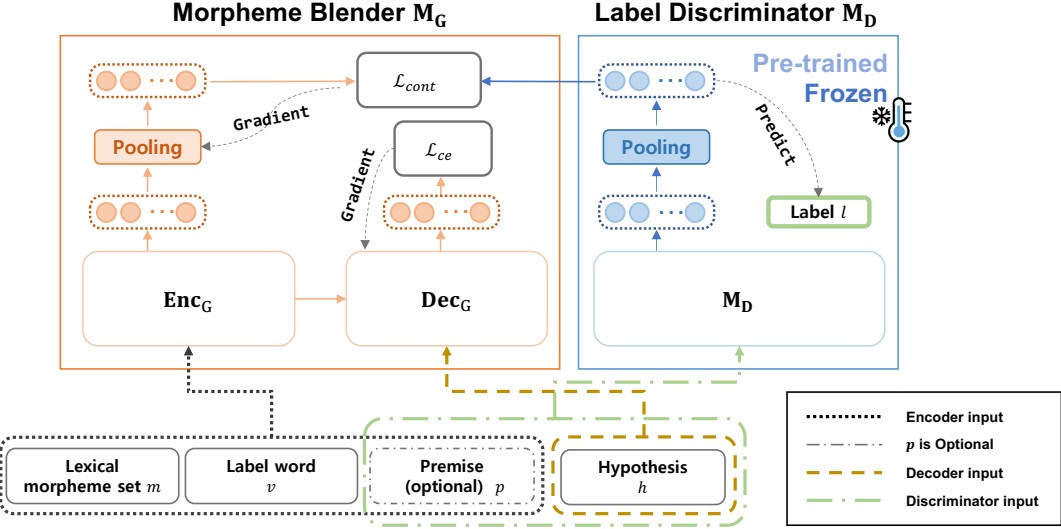

Figure 2: Training process of CHEF with Morpheme Blender $M_G$ and Label Discriminator $M_D$.

and Schütze, 2021; Zhou et al., 2022; Mekala et al., 2022). Among them, the data augmentation technique through counterfactual is particularly noteworthy, demonstrating improvements to the state-of-the-art performance levels across various benchmark datasets (Liu et al., 2021; Joshi and He, 2022; Ou et al., 2022; Wen et al., 2022). With the advent of LLMs, considerable progress has been made in overcoming augmentation constraints through generative AI (Yoo et al., 2021). Thus, we proceed assuming that it is feasible to secure textual content with LLMs. Nevertheless, it is still necessary to expend financial and temporal resources to identify the optimal direction for augmentation. As a result, we focus on determining which augmentation method could benefit model learning from Korean data. Given the diverse derivational and inflectional forms inherent in the agglutinative Korean language, we propose CHEF - an augmentation method that considers these linguistic properties.

## 3 CHEF

Taking inspiration from CommonGen (Lin et al., 2020) and Korean CommonGen (Seo et al., 2022), we focus on the data-to-text generation capabilities of sequence-to-sequence models. Common-Gen and Korean CommonGen necessitate generative commonsense reasoning based on given concept sets, where given concept sets are used to determine the feasibility of constructing plausible sentences. Both conditional generation tasks are to make generative language models learn a function $f : \mathcal{C} \rightarrow \mathcal{T}$ that maps a set of input concept

set $\mathcal{C} = \{c_1, ..., c_n\}$ to make a target sentence $\mathcal{T}$ based on the relation within $\mathcal{C}$. We perceive a parallel between this process and combining ingredients to create a complete dish. In particular, Seo et al. (2022) have shown that combining lexical morphemes yields optimal parsing for generating Korean sentences. Consequently, we concentrate on harnessing the sentence augmentation effect of sequence-to-sequence models by assembling ingredients of morpheme sets.

### 3.1 Preliminary

The primary objective of CHEF is to augment dataset $\mathcal{D} = \{(\texttt{inp}^i, l^i)\}_{i=1}^N$, where $\texttt{inp}^i = (h^i, p^i)$ denotes textual input and $l^i$ is its corresponding label. Considering multiple sentence settings such as NLI and STS, we regard $h^i$ as a hypothesis and define auxiliary input $p^i$ as a premise. If the input comprises a single sentence (*e.g.*, YNAT), we regard $h^i$ as the same text as $\texttt{inp}^i$, and $p^i$ is conceived as an empty string.

The structure of CHEF is illustrated in Figure 2. It consists of two components: the label discriminator module ($M_D$) that helps maintain label consistency and the morpheme blender ($M_G$) that augments data by referring each morpheme set. In this framework, the label discriminator module is presented as an encoder architecture, such as BERT (Devlin et al., 2018), and the morpheme blender module possesses an encoder-decoder architecture, such as BART (Lewis et al., 2020)[2].

---

[2]We adopt Korean pre-trained language models: KoBART (https://github.com/SKT-AI/KoBART) and KLUE-BERT (https://github.com/KLUE-benchmark/KLUE)

Both modules are trained to generate synthetic data according to the task and label to be augmented. In the following sections, we describe each module in detail.

## 3.2 Label Discriminator

Label discriminator module ($M_D$) helps retain the intended label consistency during the data augmentation process. $M_D$ is designed to take $inp^i$ and offer probabilities to be classified into each label. $M_D$ encode $inp$ into the vector space by applying a linear pooling layer followed by the softmax, to the [BOS] position of the last hidden state. We denote the processed output as $M_D(inp^i) \in \mathbb{R}^{1 \times n_c}$, where $n_c$ indicates the number of classes for the task dealt with $\mathcal{D}$. $M_D$ is supervised to maximize the probability of each $inp^i$ to be classified into $l^i$.

Concisely, $M_D$ is a task module trained with $\mathcal{D}$ that provides the label expectation probability obtained by the full contextualized sentence $inp^i$, and is utilized as a label instructor of the morpheme blender, described in the latter sections.

## 3.3 Morpheme Blender

**Ingredients Preparation** The main objective of the morpheme blender ($M_G$) is to synthesize the sentence by given morpheme ingredients and its targeting label. For training $M_G$, we extract lexical morpheme ingredients $m^i = \{m_1^i, \cdots, m_{n_i}^i\}$ from the $h^i$, where $n^i$ denotes the number of lexical morphemes in $h^i$. In preparing $m^i$, we use a Korean morphological analyzer, mecab-ko[3], to make the lexical morpheme set for $h^i$. We leverage that the diversity of Korean sentences that can be combined for the same lexical morpheme ingredients is potentially high. Therefore, we construct the morpheme set based on substantial lexical morphemes, allowing the morphemes within the model-aware recipe to be combined with various endings, functional morphemes, and particles.

We also define a label mapping function $\psi : l \to v$ that maps label $l^i$ into the label word $v^i$ that has a form of natural language. For instance, in the case of NLI, we adopt "contradiction" for the label "-1", "neutral" for "0", and "entailment" for "1".

In gathering these, we establish the blender that takes morpheme ingredients $m^i$ (optionally with $p^i$) and label word $v^i$ as inputs, and returns newly synthesized sentence $h'^i$. Detailed training procedures are described in the following section.

---

**Teacher Forcing** For attaining ingredient synthesizing capacity, $M_G$ is trained to generate $h^i$ by referring to the concatenated sequence of $m^i$, $p^i$, and $v^i$. The encoder framework of $M_G$, denoted as $\mathbf{Enc}_G$, receives a concatenated sequence and generates a contextualized representation by capturing the bidirectional interactions of the words in the input. Subsequently, $M_G$ produces sentences by utilizing the contextual representation acquired by the encoder. We can define the loss function $\mathcal{L}_{ce}$ for training sequence-to-sequence generation of $M_G$ as in equation (1). For clarity in expression, we denote $[\cdots]$ as a sequentialized concatenation of all the elements in it, and define the concatenated input $seq^i = [m^i, p^i, v^i]$.

$$\mathcal{L}_{ce} = -\frac{1}{|\mathcal{D}|} \sum_{i=1}^{N} \sum_j \log P_{M_G}(h_j^i \mid h_{<j}^i; seq^i) \quad (1)$$

Through the sequence-to-sequence training, $M_G$ can generate a whole sentence that covers a given morpheme set with weakly reflecting the label information granted by $v^i$ in $seq^i$.

**Label Consistency Supervision** Even if label information is reflected in the generation process by feeding it as an input, we find that generation outputs from $M_G$ still suffer from label inconsistency. To alleviate this, we exploit the auxiliary training objective utilizing pre-trained discriminator $M_G$ to consider label consistency.

We argue that by aligning the encoded outputs of $M_D$ and $\mathbf{Enc}_G$, we can make $\mathbf{Enc}_G$ better embed contextual relations with the corresponding morpheme sets and task-specific labels and promote label maintenance. This can be distilled to the training objective that maximizes the similarity $sim^i$ defined as the following equations:

$$r_G^i = \mathbf{SoftMax}(\mathbf{Enc}_G(seq^i) \cdot W_G) \quad (2)$$

$$sim^i(inp) = \frac{r_G^i \cdot M_D(inp))}{\|r_G^i\| \|M_D(inp))\|} \quad (3)$$

Note that through the pre-training of $M_D$, the encoded output of $M_D$ represents the label information of the whole sentence $h^i$ that $M_G$ should generate. The auxiliary training objective aims to make the label representation encoded through the set of ingredients $seq^i$, be aligned with the label representation yielded by $h^i$.

For the direct supervision, we define contrastive sample set $\mathcal{C}^i \subset \mathcal{D}$ for each $(inp^i, l^i) \in \mathcal{D}$, and

contrastive label set $\mathbf{l}^i = \mathbf{L} \setminus \{l^i\}$ for each label $l^i$, where $\mathbf{L}$ denotes the set of all possible labels considered in $\mathcal{D}$. In comprising $\mathcal{C}^i$, we randomly extract a single sample from $\mathcal{D}$ for each label in $\mathbf{l}^i$. Then the loss function $\mathcal{L}_{cont}$ for learning label consistency is defined as equation (4)

$$\mathcal{L}^i_{cont} = -\log \frac{\exp(\text{sim}^i(\text{inp}^i)/\tau)}{\sum_{(\text{inp}^j, l^j) \in \mathcal{C}^i} \exp(\text{sim}^i(\text{inp}^j)/\tau)} \quad (4)$$

$W_G$ denotes the linear pooling layer that maps the encoded representation on the [BOS] position into the label classification probability. The temperature parameter $\tau$ controls the sharpness of the softmax distribution with larger values leading to a smoother distribution and smaller values leading to a more peaked distribution.

In summing these, $\mathbf{M}_G$ is trained with $\mathcal{L}_{total}$ defined as the following equation:

$$\mathcal{L}_{cont} = -\frac{1}{|\mathcal{D}|} \sum_{i=1}^{N} \mathcal{L}^i_{cont} \quad (5)$$

$$\mathcal{L}_{total} = (1 - \lambda)\mathcal{L}_{ce} + \lambda \mathcal{L}_{cont} \quad (6)$$

In equation (5) and (6), $\lambda$ is the balance parameter between the two losses.

### 3.4 Augmentation Pipeline

In utilizing $\mathbf{M}_G$, we generate a single data for each $(\text{inp}^i, l^i)$ in $\mathcal{D}$. The pipeline is as follows:

1. Extract morpheme set $m^i$ of $h^i$ in $\text{inp}^i$

2. Generate $h^i_{aug} = \mathbf{M}_G([m^i, p^i, v^i])$, where $v^i$ is the label word of $l^i$.

3. $h^i_{aug}$ is regarded as an augmented data for $\mathcal{D}$, which label is $l^i$.

Considering that excessive augmentation may lead to error accumulation and the following label confusion, we apply **CHEF** to the small fraction of $\mathcal{D}$ in implementing full-shot learning.

## 4 Experimental Settings

We introduce experimental settings used for the experiments. More details are in Appendix A

### 4.1 Datasets

We adopt Korean multiple classification benchmark datasets. Each dataset is used for training and evaluation according to the proposed task format.

**KLUE-NLI** The KLUE-NLI dataset (Park et al., 2021b) has explicitly been curated for the natural language inference (NLI) task (Bowman et al., 2015). The training, validation, and test data comprises 24,998, 3,000, and 3,000 sentence pairs. In this task, models are required to process pairs of sentences, referred to as the premise and hypothesis, and deduce the underlying relationship, which could be entailment, contradiction, or neutral.

**KorNLI** The KorNLI dataset (Ham et al., 2020) is also designed for Korean natural language inference. It is generated by translating the English Standard NLI (SNLI) and Multi-Genre NLI (MNLI) datasets, as well as the Cross-lingual NLI (XNLI) dataset into Korean. The training data of the KorNLI dataset consists of 942,854 sentence pairs, machine-translated from the SNLI and MNLI datasets, while the evaluation data comprise 7,500 translated sentence pairs from the XNLI dataset.

**KLUE-STS** The KLUE-STS dataset (Park et al., 2021b) has been meticulously assembled for the semantic textual similarity (STS) task, encompassing 11,668 sentence pairs for training, 519 for validation, and 1,037 for testing. In this task, models assess pairs of sentences and determine their degree of semantic similarity.

**KLUE-YNAT** KLUE-YNAT dataset (Park et al., 2021b) has been designed for the topic classification task. The dataset includes training, validation, and test data composed of 45,678, 9,107, and 9,107 samples. In this task, models are tasked with processing sentences and assigning them to predefined news categories based on the underlying topic.

**NSMC** The NSMC dataset[4] has been constructed for the sentiment analysis task in Korean. It is derived from movie reviews and their respective ratings from the NAVER platform. The training data of the NSMC dataset comprises 150,000 reviews, and the test set consists of 50,000 reviews. In this task, models are required to process individual sentences and determine their underlying sentiment, which can be positive or negative.

### 4.2 Models

The morpheme blender employs an encoder-decoder structure and uses KoBART (Lewis et al., 2020), a pre-trained generative language model for

---

[4]https://github.com/e9t/nsmc

| Model | KLUE-NLI (Acc.) | KorNLI-MNLI (Acc.) | KorNLI-SNLI (Acc.) | KLUE-STS (Pearsonr) | KLUE-YNAT (F1) | NSMC (Acc.) |
|---|---|---|---|---|---|---|
| BERT$^K_{Base}$ | 81.53 | 80.63 | 71.56 | 90.85 | 85.73 | 90.30 |
| +KoEDA | 81.50 | 80.12 | 69.78 | 90.80 | 86.01 | 90.42 |
| +AEDA | 81.47 | 80.76 | 69.52 | 90.86 | 86.35 | 90.35 |
| +BT | 81.30 | 80.08 | 70.71 | 90.65 | 85.15 | 90.35 |
| +Mixup | 82.10 | 80.14 | 71.74 | 90.97 | 86.28 | 90.40 |
| +GPT-3.5+CA | 78.90 | 79.87 | 68.52 | 89.49 | 84.36 | 89.98 |
| +GPT-3.5+PARA | 80.40 | 80.00 | 69.42 | 91.24 | 86.32 | 90.45 |
| +GPT-4+CA | 80.73 | 80.60 | 65.26 | 90.92 | 85.50 | 90.45 |
| +GPT-4+PARA | 80.90 | 80.18 | 68.59 | **91.44** | 86.14 | 90.43 |
| **+CHEF** | **82.63** | **81.07** | **72.06** | 91.26 | **86.68** | **90.52** |
| RoBERTa$^K_{Base}$ | 84.83 | 80.84 | 71.84 | 92.50 | 85.07 | 90.08 |
| +KoEDA | 85.23 | 81.27 | 70.49 | 92.94 | 85.18 | 90.71 |
| +AEDA | 85.57 | 80.56 | 71.70 | 92.80 | 85.19 | 90.63 |
| +BT | 84.53 | 80.78 | 69.78 | 92.68 | 84.47 | 90.65 |
| +Mixup | 85.83 | 80.60 | 70.85 | 92.77 | 85.55 | 90.00 |
| +GPT-3.5+CA | 84.90 | 81.11 | 69.72 | 91.16 | 83.93 | 90.46 |
| +GPT-3.5+PARA | 84.60 | 80.58 | 70.12 | 92.95 | 85.11 | 90.87 |
| +GPT-4+CA | 84.83 | **81.85** | 62.13 | 92.91 | 84.73 | 90.83 |
| +GPT-4+PARA | 85.03 | 81.23 | 70.08 | 92.99 | 85.47 | **90.94** |
| **+CHEF** | **86.00** | 81.77 | **72.22** | **93.05** | **86.18** | 90.74 |
| RoBERTa$^K_{Large}$ | 89.17 | 81.73 | 73.26 | 93.35 | 85.69 | 91.28 |
| +KoEDA | 89.63 | 82.24 | 72.52 | 93.24 | 85.32 | 91.12 |
| +AEDA | 89.17 | 81.45 | 72.26 | 93.47 | 85.56 | 91.28 |
| +BT | 33.33 | 80.36 | 72.64 | 92.89 | 85.24 | 91.05 |
| +Mixup | **90.83** | 81.95 | 72.39 | 93.43 | 85.99 | 90.33 |
| +GPT-3.5+CA | 88.03 | 82.96 | 33.33 | 90.73 | 84.23 | 33.72 |
| +GPT-3.5+PARA | 88.33 | 83.12 | 71.86 | 93.28 | 85.88 | 91.31 |
| +GPT-4+CA | 89.10 | **83.48** | 68.94 | 93.20 | 85.46 | 67.97 |
| +GPT-4+PARA | 90.13 | 82.88 | 72.87 | 93.51 | **86.19** | 91.21 |
| **+CHEF** | 90.47 | 82.76 | **73.56** | **93.61** | 86.12 | **91.38** |

Table 1: Full-shot evaluation results on multiple classification datasets. The best performance in each dataset is **bolded**. Underlines indicate that CHEF has outperformed the baselines. Scores in red denote a training failure due to poor quality. Accuracy (Acc.), F1-score (F1), and Pearson correlation (Pearsonr) are used as evaluation metrics.

Korean, as its backbone. We choose KoBART because it exhibits acceptable Korean text generation abilities even in its small model parameters (124M) (Seo et al., 2022). KoBART takes an input sequence consisting of a lexical morpheme set, a label word, and an optional premise to generate synthetic data. The label discriminator features an encoder architecture and utilizes KLUE-BERT-base (Devlin et al., 2018), a pre-trained language model for Korean, as its backbone. KLUE-BERT-base conveys the original sentence's label to KoBART through a contrastive loss. We opt for the KLUE-BERT-base as it is the most compact model among the evaluated alternatives, mitigating the possible distillation effects that could emerge during the discrimination process due to model sizes (Park et al., 2021b). To validate the efficacy of CHEF, we select the following Korean language understanding models: BERT$^K_{Base}$ (KLUE-BERT-base), RoBERTa$^K_{Base}$ (KLUE-RoBERTa-base), and RoBERTa$^K_{Large}$ (KLUE-RoBERTa-large).

### 4.3 Compared Methods

The experiments are conducted based on full- and few-shot learning. We conduct comparative experiments using BackTranslation (**BT**) (Sennrich et al., 2016), Korean-EDA (**KoEDA**)[5](Wei and Zou, 2019), **AEDA** (Karimi et al., 2021), and **Mixup** (Zhang et al., 2018; Sun et al., 2020). We also employ **GPT-3.5** (Ouyang et al., 2022; Brown et al., 2020) and **GPT-4** (OpenAI, 2023) as backbone augmentation models. Our LLM-based approaches included semantic-based paraphrasing (**PARA**) (Fadaee et al., 2017; Kobayashi, 2018) and counterfactual augmentation (**CA**) (Liu et al., 2021; Ou et al., 2022).

## 5 Experimental Results

In this section, we evaluate a set of multiple classification datasets with our proposed method.

### 5.1 Full-shot Learning

As shown in Table 1, we compare the effectiveness and extent of performance improvement of CHEF against other data augmentation methods in the full-shot settings. We proportionally augmented the data with respect to the size of the training dataset, ensuring that an amount corresponding to 1% of the training data is added to each label.

---

[5](https://github.com/toriving/KoEDA)

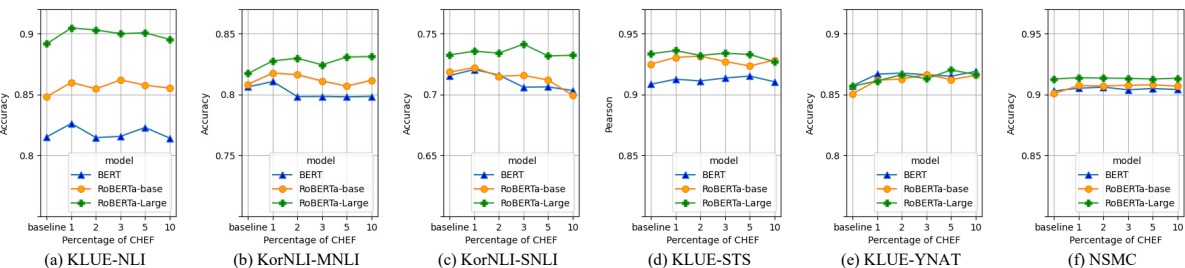

Figure 3: Performance of the language models trained on multiple classification datasets by adjusting the augmentation ratio of CHEF. The first column of each plot is the baseline performance without data augmentation.

Augmentation methods employing LLMs exhibit the generation of sentences with high diversity and superior qualitative properties. However, they occasionally demonstrate decreases in performance of up to 2%. These declines can be attributed to the significant alteration of the overall data distribution, which makes it challenging to fully account for the underlying labeling scheme intrinsic to the task. In the case of counterfactual augmentation, the instability in label transformations is amplified, even when provided with chain-of-thought prompts (Wei et al., 2022). Original sentences and labels undergo counterfactual changes without any review for conformance, resulting in increased variability that hinders the effectiveness of the augmentation process.

BT cannot guarantee the preservation of the same labels, resulting in performance decrease cases. AEDA does not show the same level of improvements for all tasks as reported in (Karimi et al., 2021) when applied to pre-trained Korean language models. Furthermore, Mixup exhibits notable performance improvements in KLUE-NLI; however, these enhancements do not consistently ensure improved results across other tasks. KoEDA utilizes a morphological analyzer to segment the data and leverages a thesaurus for Korean. This approach shows modest performance improvements, validating the effectiveness of morpheme-based data augmentation in line with the characteristics of the Korean language. However, its performance enhancements are limited due to challenges in ensuring label consistency and the absence of contextual understanding. We observe that CHEF exhibits the most significant performance improvements. By leveraging a comprehensive understanding of derivation and inflectional properties in conjunction with functional morphemes, CHEF effectively enhances models across various scenarios. As a result, CHEF overcomes these challenges and demonstrates the efficacy of data augmentation for full-shot learning.

## 5.2 Changes in Augmentation Size

Figure 3 presents the performance changes according to each dataset's augmentation ratio of CHEF. As the amount of synthetic data increases, the probability of the involvement of data that renders negative noise accumulations. In the full-shot setting, simply expanding the number of synthetic data samples does not guarantee a performance improvement. However, CHEF maintains the effectiveness of data augmentation and shows superior enhancements compared to the baseline in most cases where data augmentation is limited to within 10% (More details in Appendix 7).

## 5.3 Few-shot and Synthetic-only

We further probe the efficacy of CHEF in the NLI task, which is relatively capricious data augmentation effects in Section §5.2. We conduct comparative experiments in a few-shot setting with 32 samples and add 32 augmented sentences for each label. Table 2 shows the results averaged and maximum value over three different seeds. The data augmentation methods using the LLMs are more effective in low-resource regimes than in full-shot settings. CHEF significantly boosts the model's performance, even in situations with limited data. Furthermore, we evaluate the effectiveness of solely using the synthetic data generated by CHEF (*i.e.*, **CHEF-SynOnly**) to train the models. The outcomes reveal a spectrum of enhancements in performance and demonstrate the quality of the CHEF augmentation.

## 5.4 Ablation Study

To precisely evaluate the importance of labeling consistency, we conduct ablation studies by systematically altering the label discriminator com-

| Model | KLUE-NLI (Acc.) | KorNLI-MNLI (Acc.) | KorNLI-SNLI (Acc.) |
|---|---|---|---|
| $BERT^K_{Base}$ | 37.6/40.1 | 40.5/41.7 | 37.5/40.0 |
| +KoEDA | 43.0/45.0 | 43.5/44.9 | 36.8/38.6 |
| +AEDA | 43.1/45.5 | 42.1/43.9 | 38.0/39.8 |
| +BT | 41.1/43.0 | 42.9/44.8 | 38.0/40.2 |
| +Mixup | 40.6/41.5 | 42.8/44.2 | 38.9/39.2 |
| +GPT-3.5+CA | 37.1/37.7 | 41.6/44.6 | 38.2/39.5 |
| +GPT-3.5+PARA | 41.5/43.4 | 42.9/44.3 | 38.7/39.7 |
| +GPT-4+CA | 41.7/43.0 | 46.3/48.5 | 40.5/41.8 |
| +GPT-4+PARA | 44.7/46.1 | 43.5/45.7 | 37.7/40.7 |
| **+CHEF** | **46.5/47.6** | **47.7/49.2** | **41.1/42.3** |
| **+CHEF-SynOnly** | 41.7/42.5 | 46.8/47.4 | 35.1/37.1 |
| $RoBERTa^K_{Base}$ | 37.6/40.1 | 36.5/36.7 | 36.1/36.7 |
| +KoEDA | 41.2/44.4 | 38.4/39.9 | 36.3/36.5 |
| +AEDA | 42.4/48.0 | 39.5/42.9 | 36.8/37.2 |
| +BT | 39.9/42.2 | 38.9/41.3 | 36.2/37.1 |
| +Mixup | 39.4/41.4 | 38.4/39.5 | 36.0/36.4 |
| +GPT-3.5+CA | 36.1/36.9 | 39.1/42.6 | 36.1/37.1 |
| +GPT-3.5+PARA | 39.8/41.8 | 39.4/40.7 | 36.1/36.7 |
| +GPT-4+CA | 40.3/42.3 | 41.0/43.3 | 39.6/41.4 |
| +GPT-4+PARA | **43.3/45.8** | 39.4/41.7 | 39.4/41.6 |
| **+CHEF** | 41.2/43.3 | **43.9/45.8** | **40.1/44.7** |
| **+CHEF-SynOnly** | 38.7/39.5 | 43.0/45.1 | 33.9/34.8 |
| $RoBERTa^K_{Large}$ | 47.7/51.7 | 40.8/44.4 | 37.5/39.4 |
| +KoEDA | 51.2/52.6 | 43.6/47.0 | 38.9/40.9 |
| +AEDA | 54.6/56.1 | 45.0/48.6 | 38.7/40.3 |
| +BT | 54.4/56.3 | 43.5/48.7 | 38.0/38.8 |
| +Mixup | **57.1**/57.7 | 42.2/44.5 | 40.2/40.5 |
| +GPT-3.5+CA | 41.4/43.9 | 41.7/45.9 | 38.5/40.6 |
| +GPT-3.5+PARA | 52.6/53.7 | 43.8/47.1 | 38.5/39.8 |
| +GPT-4+CA | 45.6/51.0 | 46.1/48.8 | 41.4/44.3 |
| +GPT-4+PARA | 56.9/**59.5** | 45.3/48.2 | 38.6/41.2 |
| **+CHEF** | 56.5/57.2 | **51.5/54.5** | **44.9/47.6** |
| **+CHEF-SynOnly** | 50.0/57.1 | 49.8/54.5 | 33.8/34.2 |

Table 2: Few-shot evaluation results. The scores on the left side of the '/' represent the averages, and those on the right indicate the maximums. The best performance in each dataset is **bolded**. Underlines indicate that CHEF has outperformed the baselines.

| Model | KLUE-NLI (Acc.) | KLUE-STS (Pearsonr) | KLUE-YNAT (F1) |
|---|---|---|---|
| $BERT^K_{Base}$ | 81.53 | 90.85 | 85.73 |
| $+M_G$ | 81.83 | 90.62 | 86.02 |
| $+M_G+M_D$ | 82.54 | 91.05 | 86.34 |
| $+M_G+M_D+PT$ | **82.63** | **91.26** | **86.68** |

Table 3: Results of the ablation study on the CHEF. $M_G$ and $M_D$ respectively denote the morpheme blender and discriminator. **PT** refers to the pre-training of $M_D$. The values in **bold** represent the highest performance.

| Model | TRAIN → TEST | |
|---|---|---|
| | KLUE-NLI → Kor-NLI | Kor-NLI → KLUE-NLI |
| $BERT^K_{Base}$(Full) | **61.14** | 71.37 |
| +CHEF | 60.09 | **72.43** |
| $BERT^K_{Base}$(Few) | 41.01 | 36.06 |
| -CROSS[†] | 39.04 | 38.93 |
| +CHEF | **49.66** | **40.93** |

Table 4: Results of cross-domain analysis. → indicates that the training and test data belong to different domains. -CROSS[†] denotes the results without cross-domain settings. The best performance is **bolded**.

ponent within the CHEF. The variations in performance resulting from the exclusion or absence of pre-training in the discriminator are presented in Table 3. Upon removal of the label discriminator, we observe a decline in the augmentation effectiveness in the full-shot setting. Notably, even without the pre-training phase of the discriminator, CHEF maintains its capacity for enhancing performance. By integrating the morpheme blender and the pre-trained label discriminator using contrastive loss, CHEF achieves the highest level of performance. These empirical findings provide compelling evidence for the effectiveness of incorporating the discriminator to ensure label consistency.

## 5.5 Cross-domain Analysis

We conduct cross-domain experiments in the NLI task. As described in §4.1, KLUE-NLI and Kor-MNLI/SNLI are datasets derived from different sources. To assess performance, we employ cross-

evaluation of the model trained on data augmented using CHEF. Table 4 presents the observed performance improvements in three out of four cases, compared to the baselines without augmentation. The cross-domain effectiveness of CHEF is more pronounced in the few-shot setting. Furthermore, few-shot CHEF outperforms models trained and evaluated solely on a single domain. These results alleviate concerns regarding the degradation of the model's generalization ability due to the augmentation effects, reaffirming its robust performance across different domains.

## 5.6 Larger and Multilingual Blender

To comprehend the implications of employing a larger generative model for CHEF, we evaluate the performance leveraging the mBART-large (Liu et al., 2020). To avoid unintended knowledge distillation effects resulting from using larger discriminator modules, we maintain the same discriminator. As presented in Table 5, the results present that using a larger generative model as the blender does not necessarily ensure higher performance, but comparable effects can be achieved as well. We further find that employing a multilingual model as the blender yields similar augmentation effects.

## 5.7 Morpheme Ingredients Variants

Table 6 shows the CHEF's performance employing different morpheme ingredients. We modify

| Model | KLUE-NLI (Acc.) | KLUE-STS (Pearsonr) | KLUE-YNAT (F1) |
|---|---|---|---|
| BERT$^K_{Base}$ | 81.53 | 90.85 | 85.73 |
| +CHEF(BART) | **82.63** | **91.26** | 86.68 |
| +CHEF(mBART) | 82.00 | **91.26** | **87.11** |

Table 5: Evaluation results on larger morpheme blender. The best performance in each dataset is **bolded**.

| Model | KLUE-NLI |
|---|---|
| BERT$^K_{Base}$ | 81.53 |
| + CHEF$_{Synonym}$ | 80.22 |
| + CHEF$_{Antonym}$ | 79.84 |

Table 6: Evaluation results on morpheme ingredients variants. CHEF$_{Synonym}$ and CHEF$_{Antonym}$ denote the methods of replacing morphemes and labels.

the lexical morphemes by substituting them with synonyms or altering them to antonyms considering labels. Replacing lexical morphemes based on antonyms or synonyms is not guaranteed to preserve the compositionality across other lexical morphemes. The involvement of non-contextualized morphemes leads to diminished generative capabilities and the generation of low-quality synthetic data, resulting in a decline in performance.

## 6 Conclusion

We introduce CHEF, a novel data augmentation method designed for the Korean language, which is inherently agglutinative and rich in morphological variations. CHEF leverages the combinatory properties of lexical and functional morphemes in Korean to construct linguistically diverse and label-consistent synthetic data. By incorporating a morpheme blender and a label discriminator module, CHEF ensures that the generated synthetic data preserves the label information of the original dataset while introducing new linguistic forms through morphological alterations. Our experiments demonstrate the effectiveness and robustness of CHEF across various scenarios. In future work, we plan to explore the adaptability of the CHEF architecture to other morphologically rich languages and further optimize the interaction between the morpheme blender and label discriminator.

## Limitations

This study proposes an effective augmentation method suitable for Korean datasets, considering the unique linguistic characteristics of the Korean language. However, the method of this study primarily focuses on Korean and does not sufficiently consider other languages. This area can be further explored and improved in future research. Also, due to the performance limitations of the off-the-shelf pre-trained generative model, unnecessary word duplication occurs in sentences augmented using CHEF. This issue can be addressed by introducing a more optimized decoding strategy or employing a more advanced generative model, expected to produce higher-quality results. Moreover, The mecab-ko analyzer, which we used for constructing the set of morphemes, can have an error rate depending on the domain and data it is applied to. The proportion of morpheme combination rules in CHEF may differ based on which morpheme analyzer is used to construct the morpheme set. This presents a potential risk: if a less proficient morpheme analyzer is used, it may fail to recover the original morphemes accurately and be susceptible to errors when processing data from untrained domains. Therefore, leveraging a more advanced morpheme analyzer could enable data augmentation that more accurately reflects the linguistic characteristics of the Korean language.

## Ethics Statement

We employed Korean multiple classification benchmark datasets in our experiments. Data augmentation was conducted by altering the morphological composition of sentences present in the training datasets. Excluding the NSMC dataset, each benchmark dataset has been officially released and has undergone validation to ensure ethical considerations using human annotators. The NSMC dataset may contain some unethical expressions among negative reviews. Furthermore, we acknowledge that the pre-trained language models (PLMs), used as the backbone for the morpheme blender, could have been exposed to toxic data during the pre-training process, thereby possessing the potential to generate inappropriate synthetic data.

## Acknowledgments

This research was supported by the MSIT (Ministry of Science and ICT), Korea, under the ITRC (Information Technology Research Center) support program (IITP-2023-2018-0-01405) supervised by the IITP (Institute for Information & Communications Technology Planning & Evaluation). This work was supported by Institute of Information &

communications Technology Planning & Evaluation (IITP) grant funded by the Korea government (MSIT) (No. 2020-0-00368, A Neural-Symbolic Model for Knowledge Acquisition and Inference Techniques). This research was supported by the MSIT (Ministry of Science and ICT), Korea, under the ICT Creative Consilence program (IITP-2023-2020-0-01819) supervised by the IITP(Institute for Information & communications Technology Planning & Evaluation).

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

# A    Experimental Details

We trained our models on a single NVIDIA A6000 GPU (48GB) and AMD EPYC 7513 32-Core Processor CPUs.

**CHEF Modules**    As a discriminator, CHEF uses KLUE-BERT-base[6](Devlin et al., 2018; Park et al., 2021a), which has 768 embedding sizes, 768 hidden sizes, 12 layers, and 12 attention heads. As a morpheme blender, CHEF employs KoBART-base[7](Lewis et al., 2020), where each encoder and decoder have 768 hidden sizes, 6 layers, and 16 attention heads, respectively.

**Korean Language Understanding Models**    We used three pre-trained language models for Korean. KLUE-BERT-base (Devlin et al., 2018; Park et al., 2021a) has 768 embedding sizes, 768 hidden sizes, 12 layers, and 12 attention heads. KLUE-RoBERTa-base (Liu et al., 2019; Park et al., 2021a) also has 768 embedding sizes, 768 hidden sizes, 12 layers, and 12 attention heads. KLUE-RoBERTa-large (Liu et al., 2019; Park et al., 2021a) has 1024 embedding sizes, 1024 hidden sizes, 24 layers, and 16 attention heads.

**EDA & BackTranslation**    We applied Back-Translation (BT) (Sennrich et al., 2016) and easy data augmentation (EDA) (Wei and Zou, 2019) to the data to generate synthetic data. For BT, we used M2M100 (Fan et al., 2021) to translate Korean text into Japanese and back into Korean. The model we used for BT is facebook-M2M100 (418M), which has a 1024 embedding size, 12 layers, and 16 attention heads. We used KoEDA [8], a library that utilizes the Korean WordNet (Bikel, 2000) for EDA.

---

[6]https://github.com/KLUE-benchmark/KLUE
[7]https://github.com/SKT-AI/KoBART
[8]https://github.com/toriving/KoEDA

**AEDA & Mixup**  We used data augmentation with the default settings presented in the AEDA paper (Karimi et al., 2021)[9]. Mixup (Zhang et al., 2018; Sun et al., 2020), which is suitable for Transformer-based models, was implemented by randomly shuffling sample indices within a batch. Subsequently, the hidden state of each sample and the label scalar were mixed at a ratio of 0.2, following the default $\lambda$ value, to create synthetic data.

**Large Language Models**  We included GPT-3.5 (gpt-3.5-turbo-0301) (Brown et al., 2020; Ouyang et al., 2022) and GPT-4 (gpt-4-0314) (OpenAI, 2023) as the large language models. We applied augmentation to the given training data using the OpenAI API[10] and LangChain[11]. As depicted in Figure 4 and 5, the prompt consists of an example in a one-shot template for the given task and instructions for the augmentation method. The cost incurred due to the OpenAI API calls amounted to $245.17, and the data augmentation approach utilizing LLMs did not show a significant improvement in model performance relative to the cost incurred.

**Hyperparameters**  Hyperparameters to train the CHEF are batch size 8, learning rate $1 \times 10^{-5}$, AdamW optimizer (Loshchilov and Hutter, 2019) ($\beta_1 = 0.9$, $\beta_2 = 0.999$, $\epsilon = 1e - 8$), lambda 0.2, max source length 200, max target length 168, and 5 epochs. To train Korean language understanding models is batch size 32, learning rate $2 \times 10^{-5}$, AdamW optimizer ($\beta_1 = 0.9$, $\beta_2 = 0.999$, $\epsilon = 1e-8$), and 20 epochs. In the case of few-shot learning, all other hyperparameter settings remained unchanged except for the epoch adjusted to 30, considering the model's overfitting point.

**Decoding Strategy**  Within the framework of CHEF augmentation, we imposed specific constraints on the decoding strategy. We established the following settings: a beam size of 10, a maximum sequence length of 168, a minimum sequence length of 5, a repetition penalty of 2, and a no-repeat n-gram size of 3 to penalize the generation of duplicate tokens.

## B  Qualitative Analysis

We applied CHEF to several benchmark datasets and conducted qualitative analyses of the aug-mented output. CHEF augmentations typically take the following form of three types of variations.

As described in Figure 6, we observe that the outputs generated by CHEF primarily involve modifications of particles and determiners that are closely associated with nouns. However, no significant changes are observed in the verbs of the sentences. The combination of lexical morphemes introduces slight variations in the meaning of the generated sentences but does not lead to changes in the labels assigned to them.

As shown in Figure 7, the augmented output primarily consisted of conjugation variations, leading to a diverse range of sentences with different ending conjunction rules. However, there is minimal impact on the overall meaning of the sentences themselves. Notably, the augmented hypotheses are labeled as having significantly lower similarity to the corresponding premises regarding semantic similarity evaluation.

As illustrated in Figure 8, the augmentation process involves single sentences resembling news headlines fitting into the "Life & Culture" section. Unlike the other benchmark datasets, KLUE-YNAT is characterized by the fact that the original sentences contain relatively fewer functional morphemes. Based on this characteristic of the dataset, CHEF was applied to augment the data by changing the order or role of the lexical morphemes in the sentence.

---

[9] https://github.com/akkarimi/aeda_nlp
[10] https://openai.com/
[11] https://python.langchain.com/en/latest/

| Model | KLUE-NLI | KorNLI-MNLI | KorNLI-SNLI | KLUE-STS | KLUE-YNAT | NSMC |
|---|---|---|---|---|---|---|
| BERT$^K_{Base}$ | 81.53 | 80.63 | 71.56 | 90.85 | 85.73 | 90.30 |
| +CHEF 1% | **82.63** | **81.07** | **72.06** | 91.26 | 86.68 | 90.52 |
| +CHEF 2% | 81.47 | 79.83 | 71.60 | 91.12 | 86.76 | **90.61** |
| +CHEF 3% | 81.57 | 79.85 | 70.61 | 91.36 | 86.63 | 90.39 |
| +CHEF 5% | 82.30 | 79.81 | 70.63 | **91.52** | 86.51 | 90.50 |
| +CHEF 10% | 81.43 | 79.85 | 70.33 | 91.05 | **86.90** | 90.41 |
| RoBERTa$^K_{Base}$ | 84.83 | 80.84 | 71.84 | 92.50 | 85.07 | 90.08 |
| +CHEF 1% | 86.00 | **81.77** | **72.22** | 93.05 | 86.18 | 90.74 |
| +CHEF 2% | 85.47 | 81.63 | 71.50 | **93.16** | 86.28 | 90.70 |
| +CHEF 3% | **86.20** | 81.11 | 71.58 | 92.71 | **86.64** | 90.76 |
| +CHEF 5% | 85.77 | 80.70 | 71.19 | 92.35 | 86.23 | **90.80** |
| +CHEF 10% | 85.53 | 81.17 | 69.94 | 92.83 | 86.58 | 90.69 |
| RoBERTa$^K_{Large}$ | 89.17 | 81.73 | 73.26 | 93.35 | 85.69 | 91.28 |
| +CHEF 1% | **90.47** | 82.76 | 73.56 | **93.61** | 86.12 | **91.38** |
| +CHEF 2% | 90.30 | 82.98 | 73.40 | 93.21 | 86.63 | 91.35 |
| +CHEF 3% | 90.00 | 82.44 | **74.14** | 93.40 | 86.30 | 91.33 |
| +CHEF 5% | 90.07 | 83.08 | 73.17 | 93.29 | **87.02** | 91.28 |
| +CHEF 10% | 89.53 | **83.12** | 73.23 | 92.69 | 86.64 | 91.34 |

Table 7: Performance of language models on multiple classification datasets by adjusting CHEF's augmentation rate. The best-performing method on each dataset is shown in **bold**. Underlines indicate that the method outperformed the baselines.

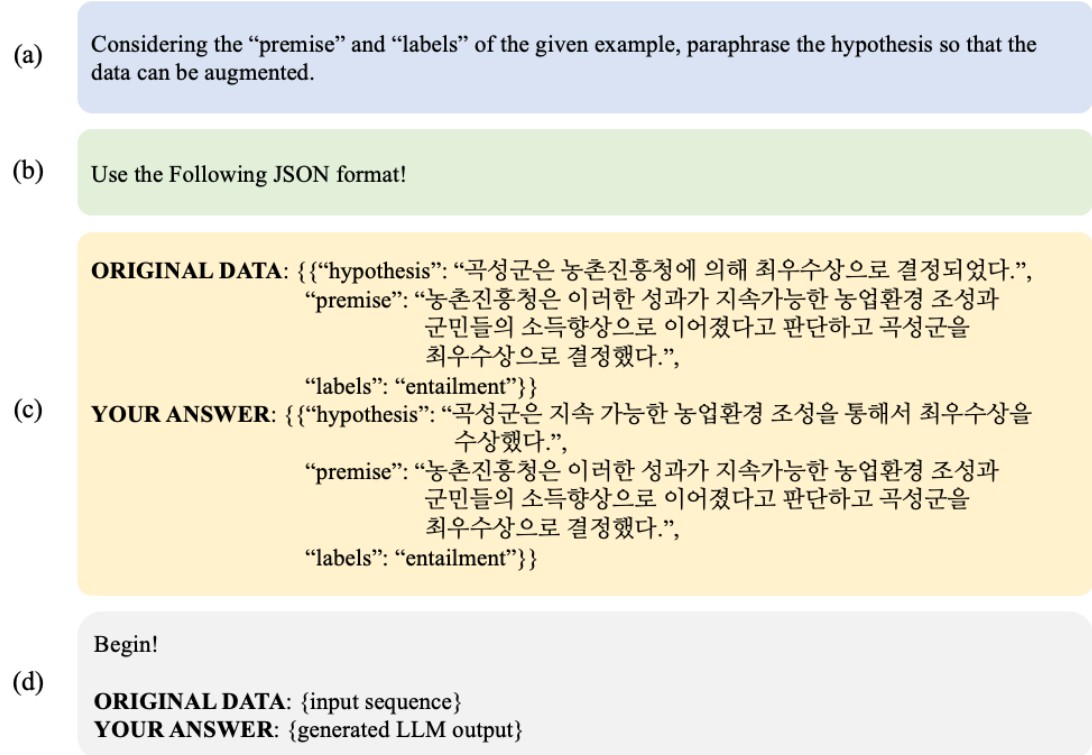

Figure 4: An example of a prompt template for augmenting data with given original data. (a) is a human instruction to paraphrase the hypothesis of the given data. (b) is an additional directive to limit the output to JSON format. (c) is an example of paraphrasing. (d) is the format of the given input sequence and the generated LLM output.

(a) Considering the "premise" and "labels" of the given example, make the counterfactual hypothesis so that the data can be augmented.

A SET OF "labels": ["entailment", "contradiction", "neutral"].

You should randomly select one of the labels in A SET OF "labels" and change the "labels" in YOUR ANSWER to be different from the "labels" in the ORIGINAL DATA and paraphrase with an appropriate "hypothesis".

(b) Use the Following JSON format!

(c) ORIGINAL DATA: {{"hypothesis": "곡성군은 농촌진흥청에 의해 최우수상으로 결정되었다.",
"premise": "농촌진흥청은 이러한 성과가 지속가능한 농업환경 조성과 군민들의 소득향상으로 이어졌다고 판단하고 곡성군을 최우수상으로 결정했다.",
"labels": "entailment"}}
YOUR ANSWER: {{"hypothesis": "농촌진흥청은 곡성군의 성과가 부족하다고 판단하여 예선 탈락시켰다.",
"premise": "농촌진흥청은 이러한 성과가 지속가능한 농업환경 조성과 군민들의 소득향상으로 이어졌다고 판단하고 곡성군을 최우수상으로 결정했다.",
"labels": "contradiction"}}

(d) Begin!

ORIGINAL DATA: {input sequence}
YOUR ANSWER: {generated LLM output}

Figure 5: An example of prompt template for augmenting data with given original data. (a) is a human instruction for generating a counterfactual sequence for a hypothesis on a given input and instructions on how to set the label of the counterfactual sequence. (b) is an additional directive to limit the output to JSON format. (c) is an example of counterfactual sequence generation. (d) is the format of the given input sequence and the generated LLM output.

| KLUE-NLI | Contradiction |

Premise: 14일 오전 인스타그램 서버 다운으로 인한 오류가 발생했다.
(There was an error due to the Instagram server downtime on the morning of the 14th day.)

Hypothesis: 14일 내내 어떠한 오류도 발생하지 않았다.
(There were no errors whatsoever throughout the 14th day.)

| CHEF Augmented | Lexical Morphemes: {14일, 발생, 않-, 오류} |

CHEF Hypothesis ① 14일은 어떠한 오류도 발생하지 않았다. (No errors occurred on the 14th day.)
CHEF Hypothesis ② 14일에도 오류는 발생하지 않았다. (No errors occurred on the 14th day as well.)
CHEF Hypothesis ③ 14일 오전 오류는 발생하지 않았다. (No errors occurred in the morning of the 14th day.)
CHEF Hypothesis ④ 오류는 발생하지 않았다. (No errors occurred.)
CHEF Hypothesis ⑤ 14일에는 전혀 오류가 발생하지 않았다. (There were no errors whatsoever on the 14th day.)

Figure 6: An example of augmentation result from CHEF in KLUE-NLI

| KLUE-STS | | Similarity 0.1 |

**Premise:** 또 겨울이 우기인 프랑스의 올씨년스런 분위기와 방 곳곳에 위치한 등 조명, 연식 있는 건물이 잘 어울렸습니다.
(The wintery atmosphere of France, which is known for its elegant ambiance, along with the scattered lighting and aged buildings throughout the rooms, blended perfectly.)

**Hypothesis:** 저의 여행일정에 위치나 가격대 시설 등이 만족스러웠습니다.
(The location, price range, and facilities of the accommodations in my travel itinerary were satisfactory.)

| CHEF Augmented | Lexical Morphemes: {여행, 일정, 위치, 가격대, 시설, 만족-} |

**CHEF Hypothesis ①** 위치, 가격대, 시설, 위치, 여행일정 등 모든 일정에 만족하였습니다.

(I was satisfied with everything, including the location, price range, facilities, and travel itinerary.)

**CHEF Hypothesis ②** 위치, 가격대, 시설, 위치, 위치 등 여행일정 일정에 만족했습니다.

(I was satisfied with the location, price range, facilities, and overall travel itinerary, including the locations visited.)

**CHEF Hypothesis ③** 위치, 여행시설, 가격대, 시설, 위치, 위치 등 모든 일정은 만족스러웠습니다.

(I was satisfied with everything, including the location, travel accommodations, price range, facilities, and overall itinerary.)

Figure 7: An example of augmentation result from CHEF in KLUE-STS

| KLUE-YNAT | | Life & Culture |

**Title:** 강풍·영하권 날씨...오후부터 비까지 내려
(Bracing for Strong Winds and Subzero Temperatures... Afternoon Rain in the Forecast)

| CHEF Augmented | Lexical Morphemes: {강풍, 영하권, 비, 오후, 내-, 날씨} |

**CHEF Hypothesis ①** 내일날씨 오후부터 기온 뚝...오후부터 강풍·오후 영하권 날씨

(Temperature Dropping Tomorrow Afternoon... Strong Winds and Subfreezing Temperatures in the Afternoon)

**CHEF Hypothesis ②** 날씨 오후 강풍·날씨 영하권 강풍...오후부터 오후까지 비

(Afternoon Gale Force Winds and Subfreezing Conditions... Rain Expected from Afternoon to Evening)

**CHEF Hypothesis ③** 내일날씨 오후부터 영하권 날씨...오후엔 강풍",

(Subfreezing Conditions Expected from Tomorrow Afternoon... Strong Winds in the Afternoon)

Figure 8: An example of augmentation result from CHEF in KLUE-YNAT