# OpenReview forum: "CHEF in the Language Kitchen: A Generative Data Augmentation Leveraging Korean Morpheme Ingredients"
_EMNLP/2023/Conference — EMNLP 2023 Main_

### Official Review · Reviewer_nopA · 2023-08-05

**Soundness:** 4

**Excitement:**

3: Ambivalent: It has merits (e.g., it reports state-of-the-art results, the idea is nice), but there are key weaknesses (e.g., it describes incremental work), and it can significantly benefit from another round of revision. However, I won't object to accepting it if my co-reviewers champion it.

**Paper Topic And Main Contributions:**

This paper proposes a data augmentation method for Korean, named CHEF, which can generate augmented samples with more diverse morphemes. Experiments show that the proposed CHEF for data augmentation can achieve the best performance on several Korean downstream tasks and outperform state-of-the-art LLMs (i.e., GPT-3.5, GPT-4).

**Reasons To Accept:**

- The proposed method is intuitive from the perspective of linguistics.
- Experiments prove the effectiveness of the proposed method on several tasks.

**Reasons To Reject:**

- There are some other data augmentation methods (Guo, H. 2020, Karimi et al., 2021) that should be included for comparison.
- It would be better if the scope of the paper is larger, e.g., testing the method on Japanese (also an agglutinative language) datasets.


1. Hongyu Guo. Nonlinear Mixup: Out-Of-Manifold Data Augmentation for Text Classification. AAAI'20.
1. Karimi et al. AEDA: An Easier Data Augmentation Technique for Text Classification. EMNLP'21 Findings.

**Reproducibility:**

4: Could mostly reproduce the results, but there may be some variation because of sample variance or minor variations in their interpretation of the protocol or method.

**Reviewer Confidence:**

3: Pretty sure, but there's a chance I missed something. Although I have a good feel for this area in general, I did not carefully check the paper's details, e.g., the math, experimental design, or novelty.

---

> ### Author Rebuttal · Authors · 2023-08-25
>
> Thanks for your careful and valuable comments.
>
> > R1: *'There are some other data augmentation methods (Guo, H. 2020, Karimi et al., 2021) that should be included for comparison'*
>
> We appreciate your suggestion and will certainly include experiments comparing with **MIXUP** and** AEDA** in our evaluation. The results with these two methods in **full-shot learning** using default settings are as follows:
>
> 1) KLUE-BERT-base
>
> |Method|KLUE-NLI|Kor-MNLI|Kor-SNLI|KLUE-STS|KLUE-YNAT|NSMC|
> |--- |--- |--- |--- |--- |--- |--- |
> |Baseline|81.53|80.63|71.56|90.85|85.73|90.30|
> |AEDA [5] |81.47|80.76|69.52|90.86|86.35|90.35|
> |MIXUP [2,4]|82.10|80.14|71.74|90.97|86.28|90.40|
> |CHEF|**82.63**|**81.07**|**72.06**|**91.26**|**86.68**|**90.52**|
>
> 2) KLUE-RoBERTa-base
>
> |Method|KLUE-NLI|Kor-MNLI|Kor-SNLI|KLUE-STS|KLUE-YNAT|NSMC|
> |--- |--- |--- |--- |--- |--- |--- |
> |Baseline|84.83|80.84|71.84|92.50|85.07|90.08|
> |AEDA [5] |85.57|80.56|71.70|92.80|85.19|90.63|
> |MIXUP [2,4]|85.83|80.60|70.85|92.77|85.55|90.00|
> |CHEF|**86.00**|**81.77**|**72.22**|**93.05**|**86.18**|**90.74**|
>
>
> 3) KLUE-RoBERTa-large
>
> |Method|KLUE-NLI|Kor-MNLI|Kor-SNLI|KLUE-STS|KLUE-YNAT|NSMC|
> |--- |--- |--- |--- |--- |--- |--- |
> |Baseline|89.17|81.73|73.26|93.35|85.69|91.28|
> |AEDA [5] |90.17|81.45|72.26|93.47|85.56|91.28|
> |MIXUP [2,4]|**90.83**|81.95|72.39|93.43|85.99|90.33|
> |CHEF|90.47|**82.76**|**73.56**|**93.61**|**86.12**|**91.38**|
>
> However, it is worth noting that **Nonlinear Mixup** [1] is primarily suited for **CNN/LSTM**-based neural network models and is known to be challenging to fully apply to Transformer-based pre-trained language models [2] [3]. **Therefore, we have used the more advanced Mixup approach [2] [4], which is compatible with pre-trained language models**, to evaluate the performance.
>
> The **two newly added baselines** demonstrate good augmentation effects but also **face challenges in improving performance on Natural Language Inference (NLI) tasks**. Furthermore, they also tend to be **unstable** in their performance like other baselines, sometimes completely **failing to converge** depending on the initial seed value, which often necessitates the assignment of a new seed for taking the results.
>
> We will include the complete experimental results in the final version of the paper.
>
> [1] Hongyu Guo. Nonlinear Mixup: Out-Of-Manifold Data Augmentation for Text Classification. AAAI'20.
>
> [2] Sun, L., Xia, C., Yin, W., Liang, T., Philip, S. Y., & He, L. (2020, December). Mixup-Transformer: Dynamic Data Augmentation for NLP Tasks. In Proceedings of the 28th International Conference on Computational Linguistics (pp. 3436-3440).
>
> [3] Yoon, S., Kim, G., & Park, K. (2021, August). SSMix: Saliency-Based Span Mixup for Text Classification. In Findings of the Association for Computational Linguistics: ACL-IJCNLP 2021 (pp. 3225-3234).
>
> [4] Kong, F., Zhang, R., Guo, X., Mensah, S., & Mao, Y. (2022, December). Dropmix: A textual data augmentation combining dropout with mixup. In Proceedings of the 2022 Conference on Empirical Methods in Natural Language Processing (pp. 890-899).
>
> [5] Karimi et al. AEDA: An Easier Data Augmentation Technique for Text Classification. EMNLP'21 Findings.
>
> > R2: *'It would be better if the scope of the paper is larger, e.g., testing the method on Japanese (also an agglutinative language) datasets.'*
>
> Thank you for the valuable suggestion. The scope of this research aimed to demonstrate the effectiveness of text augmentation by leveraging the "agglutinative" characteristics specific to the “Korean language”. **In this study, we aimed for a vertical analysis** focused on one language. **In future research**, we plan to analyze the effectiveness of "CHEF" in other agglutinative languages like Japanese, Turkish, etc., **as you suggested, to broaden the scope of our study horizontally**.

---

### Official Review · Reviewer_p68b · 2023-08-05

**Soundness:** 4

**Excitement:**

3: Ambivalent: It has merits (e.g., it reports state-of-the-art results, the idea is nice), but there are key weaknesses (e.g., it describes incremental work), and it can significantly benefit from another round of revision. However, I won't object to accepting it if my co-reviewers champion it.

**Paper Topic And Main Contributions:**

The paper presents a method for data augmentation for Korean based on generating morphological variations while preserving a sentence-level label.

**Questions For The Authors:**

Please clarify the scope of the morphological blender and the label discriminator (word or sentence level)

Please clarify the input of the morphological blender w.r.t. lexical and functional morphemes.

Please clarify the role of KoBART and KLUE-BERT in the morphological blender and the label discriminator respectively.

**Reasons To Accept:**

The paper addresses issues important for morphologically rich languages, and adds to the work on automatic data augmentation. There is extensive evaluation on a variety of datasets, and with different LLM + data augmentation options, in the full training set, using different augmentation ratios, and few-shot settings.

**Reasons To Reject:**

Several issues could be clarified:

- because the focus is on morphology, which is usually word-level, it is not clear that the morpheme blender and label discriminator are sentence-level, and not word level. It would be very useful to make this clear from the beginning (if indeed this is the case). It would also be useful to provide some concrete example label (maybe in Figure 1, instead of the abstract "+1" label)

- based on the examples in the appendix, it seems that the morphological blender receives as input only lexical morphemes. Is this the case? If yes, where do the functional morphemes come from? Or the blender is more than a blender, and more a generator? If yes, maybe the term "morphological blender" is a bit misleading.

- it would be useful to include a brief description of CommonGen, and that should help provide a more informative contrast with the method proposed here.

- Section 4.2. is not very clear. What does KoBART correspond to in the morpheme blender? What does KLUE-BERT correspond to in the label discriminator?

**Reproducibility:**

4: Could mostly reproduce the results, but there may be some variation because of sample variance or minor variations in their interpretation of the protocol or method.

**Reviewer Confidence:**

2: Willing to defend my evaluation, but it is fairly likely that I missed some details, didn't understand some central points, or can't be sure about the novelty of the work.

---

> ### Author Rebuttal · Authors · 2023-08-25
>
> Thank you so much for your comments. Please check the following regarding your concerns.
>
> > R1-1 & Q1: *'Because the focus is on morphology, which is usually word-level, it is not clear that the morpheme blender and label discriminator are sentence-level, and not word level & Please clarify the scope of the morphological blender and the label discriminator (word or sentence level)'*
>
> We **extract lexical morphemes** with substantive meaning **from the original sentence** **segmented at the morphological level** for data augmentation. The **morpheme blender attaches functional morphemes to these lexical morphemes to create sentences** of different forms but similar meanings. Meanwhile, the label discriminator ensures that the sentence generated by the morpheme blender retains the original sentence's label (e.g., +1 or -1).
>
> The confusion may arise from the unique feature of Korean, where multiple morphemes within a single word can be detached and recombined. **Our proposed method involves decomposing a single sentence into its morpheme units and then reconstructing it based on lexical morphemes.**
>
> Therefore, there is **no need to distinguish between word-level and sentence-level** in the context of our proposed augmentation.
>
> > R1-2: *'It would be very useful to make this clear from the beginning (if indeed this is the case). It would also be useful to provide some concrete example label (maybe in Figure 1, instead of the abstract "+1" label)'*
>
> We agree that a clearer explanation for Figure 1 would be beneficial. In line with the reviewer's suggestion, **we will add a concrete example to Figure 1** and **clarify the explanation about the label by revising it to include concrete labels (e.g., positive & negative).** Additionally, we will enhance the content of the caption to reflect these changes, ensuring that authors who are not familiar with Korean will not experience misunderstanding.
>
> **We will also make revisions to strengthen the content on page 2, lines 69-71 of the Introduction section. The revised text will be as follows:**
>
> *'The role of the morpheme blender is to take the concatenated sequence of morphemes and returns an intact sentence that contains given morphemes. Label discriminator aids morpheme blender to generate a label-consistent sentence by comparing encoded representations generated by the morpheme set and a whole sentence.'*
>
> > R2-1 & Q2: *'Based on the examples in the appendix, it seems that the morphological blender receives as input only lexical morphemes. Is this the case? If yes, where do the functional morphemes come from? & Please clarify the input of the morphological blender w.r.t. lexical and functional morphemes.'*
>
> **Partially Yes**. As depicted in Figure 2, the morpheme blender requires three (or two) types of inputs for the encoder-decoder architecture: **(1) a morpheme set, (2) a label word, (3) a premise (optional)** for teacher forcing and contrastive learning. **The morpheme set (1) consists only of lexical morphemes**.
>
> In accordance with the **conditional generation task**, **the morpheme blender uses the sequence-to-sequence model (we used 'KoBART'), which takes lexical morphemes as an input source and attaches functional morphemes to generate a complete sentence**. The rules for attaching functional morphemes are based on the **linguistic prior knowledge of the Korean language** as captured by the pre-trained language model, KoBART.
>
> The paper on page 4, lines 235 - 239 covers this process, where the rules for attaching functional morphemes applied by the morpheme blender are considered part of **the model-aware recipe**.
>
> For clearer elucidation, **we will specify in Figure 2 that the morpheme set consists of lexical morphemes**. Additionally, **we will revise the text on page 4, line 227**, from '*denotes the number of morphemes*' to '***denotes the number of lexical morphemes.***'
>
> > R2-2: *'Or the blender is more than a blender, and more a generator? If yes, maybe the term "morphological blender" is a bit misleading.'*
>
> **The core of CHEF lies in blending lexical and functional morphemes**. As illustrated in Figure 2, we also utilize contrastive loss to incorporate label information into the blend. To clarify that the output of the morpheme blender results in sentence generation through the decoder, we provided a detailed explanation about **“teacher forcing” in a bold paragraph on page 4, lines 250-269**. Direct explanations regarding **"sentence generation" are also given on page 4, lines 251 and 266-269**. Furthermore, through the **“Augmentation Pipeline” in Section 3.4 on page 5**, we indicated that the morpheme blender is involved in the sentence generation process.
>
> **To address the reviewer's question more clearly in the paper, we will add an explanation regarding the term "morpheme blender."** The added text will be as follows:
>
> *'Morpheme blender takes morphemes and generates a sentence that contains given morphemes. In this sense, we denote it as a blender.'*
>
> > R3: *'It would be useful to include a brief description of CommonGen, and that should help provide a more informative contrast with the method proposed here.'*
>
> We will add an explanation about CommonGen according to your advice. The additional text will be placed on page 3, line 173, as follows:
>
> '*Both conditional generation tasks use a set of input concept set* $\mathcal{C}$, *where* $\{ c_1, c_2, \ldots, c_m \}$ *to make a target sentence* $\mathcal{T}$, *where* $\mathcal{T} = \{ t_1, t_2, \ldots, t_n \}$.
> *Generative language models for those tasks learn a function* $\mathcal{f}: \mathcal{X} \rightarrow \mathcal{Y}$ *that maps* $\mathcal{C}$ *to* $\mathcal{T}$'.
>
> > R4 & Q3: *'Section 4.2. is not very clear. What does KoBART correspond to in the morpheme blender? What does KLUE-BERT correspond to in the label discriminator? & Please clarify the role of KoBART and KLUE-BERT in the morphological blender and the label discriminator, respectively.'*
>
> **As described in Page 3, lines 191-202**, KoBART is the backbone model for the morpheme blender and features an encoder-decoder architecture. On the other hand, KLUE-BERT-base, an auto-encoder architecture, functions as the backbone for the label discriminator.
>
> **We appreciate the reviewer's comments and will clarify the description in Section 4.2 as follows**:
>
> *'The morpheme blender employs an encoder-decoder structure and uses KoBART, a pre-trained generative language model for Korean, as its backbone. KoBART takes an input sequence consisting of a lexical morpheme set, a label word, and an optional premise to generate synthetic data. The label discriminator features an encoder architecture and utilizes KLUE-BERT-base, a pre-trained language model for Korean, as its backbone. KLUE-BERT-base conveys the original sentence's label to KoBART through a contrastive loss.'*

---

### Official Review · Reviewer_ACJB · 2023-08-05

**Soundness:** 4

**Excitement:**

4: Strong: This paper deepens the understanding of some phenomenon or lowers the barriers to an existing research direction.

**Paper Topic And Main Contributions:**

This paper proposes an efficient data augmentation method for Korean language by generating diverse sentence forms using lexical and functional morphemes. The architecture is based on a generator-discriminator model for a sentence generation and its label consistency. This method is evaluated with the KLUE benchmark and showed its efficiency.

**Reasons To Accept:**

The method showed significant improvement in the KLUE benchmark test even though it uses relatively smaller language model (BERT and BART) than GPT4. The paper is easy to follow and the extensive experiments support the proposed method.

**Reasons To Reject:**

This paper does not appropriately discuss the linguistic issues on irregular changes in Korean language generation: this paper uses mecab-ko analyzer to extract morpheme ingredients, but the analyzer does not provide linguistically correct Korean morphemes, failing to recover original morphemes in irregular verbs and adjectives. More explanation is needed for this choice and its impact on the performance.

**Reproducibility:**

4: Could mostly reproduce the results, but there may be some variation because of sample variance or minor variations in their interpretation of the protocol or method.

**Reviewer Confidence:**

3: Pretty sure, but there's a chance I missed something. Although I have a good feel for this area in general, I did not carefully check the paper's details, e.g., the math, experimental design, or novelty.

---

> ### Author Rebuttal · Authors · 2023-08-25
>
> We appreciate your in-depth points.
>
> > R1-1: *'This paper uses mecab-ko analyzer to extract morpheme ingredients, but the analyzer does not provide linguistically correct Korean morphemes, failing to recover original morphemes in irregular verbs and adjectives. More explanation is needed for this choice and its impact on the performance.'*
>
> **The mecab-ko analyzer is capable of restoring the original form of Korean morphemes**. This analyzer may not always perform perfectly, having some error rate depending on the domain and the data. **However**, the issue raised by the reviewer regarding the *'failing to recover original morphemes in irregular verbs and adjectives'* **is almost negligible when using mecab-ko analyzer with CHEF**. **In cases where the irregular form should be applied, CHEF can correctly apply the irregular change**, and likewise, in cases where the regular form should be applied, CHEF can correctly apply the regular change.
>
> For example, in the KLUE-STS dataset augmented by CHEF, instances of the 'ㄷ' irregular change and regular change both appeared.
>
> ex) Synthetic 1: KOR: … 더 많이 묻는… / ENG: ... asking more … (‘ㄷ' irregular change)
>
> Synthetic 2: KOR: … 더 많이 물어보는… / ENG: ... asking more … (regular change)
>
> The majority of the synthesized sentences were **mapped according to regular/irregular changes**, with lexical morphemes correctly aligned with functional morphemes.
>
> To elaborate further, the mecab-ko analyzer has been proven effective in restoring the original form of Korean morphemes in various research and posts. **It is a well-recognized open-source tool that has significantly contributed to Korean morphological analysis research.** We do not believe that the following studies, which employ this analyzer, are using an incorrect tool for Korean morpheme analysis [1] [2] [3] [4].
>
> [1] *Park, K., Lee, J., Jang, S., & Jung, D. (2020, December). An Empirical Study of Tokenization Strategies for Various Korean NLP Tasks. In Proceedings of the 1st Conference of the Asia-Pacific Chapter of the Association for Computational Linguistics and the 10th International Joint Conference on Natural Language Processing (pp. 133-142).*
>
> [2] *Park, C., Eo, S., Moon, H., & Lim, H. S. (2021, June). Should we find another model?: Improving neural machine translation performance with ONE-piece tokenization method without model modification. In Proceedings of the 2021 Conference of the North American Chapter of the Association for Computational Linguistics: Human Language Technologies: Industry Papers (pp. 97-104).*
>
> [3] *Moon, S., Cho, W. I., Han, H. J., Okazaki, N., & Kim, N. S. (2022, June). OpenKorPOS: Democratizing Korean Tokenization with Voting-Based Open Corpus Annotation. In Proceedings of the Thirteenth Language Resources and Evaluation Conference (pp. 4975-4983).*
>
> [4] *Kim, A., & Kim, J. (2022, May). Vacillating Human Correlation of SacreBLEU in Unprotected Languages. In Proceedings of the 2nd Workshop on Human Evaluation of NLP Systems (HumEval) (pp. 1-15).*
>
> We also aimed to use the **morpheme analyzer that has previously demonstrated its effectiveness**, taking into account the performance of morphological analyzers can be significantly influenced by the data or domain in which they are used. **The Korean text-generation research by Seo et al. 2022 [5] has demonstrated** that when mecab-ko analyzer is combined with a generative language model, it can **achieve performance close to the upper bound of human-level capability**. This proves that the **mecab-ko analyzer can properly handle both irregular and regular morphological rules** and can effectively reconstruct sentences when paired with a generative language model.
>
> [5] *Seo, J., Lee, S., Park, C., Jang, Y., Moon, H., Eo, S., ... & Lim, H. S. (2022, July). A dog is passing over the jet? a text-generation dataset for korean commonsense reasoning and evaluation. In Findings of the Association for Computational Linguistics: NAACL 2022 (pp. 2233-2249).*
>
>
> > R1-2: *‘This paper does not appropriately discuss the linguistic issues on irregular changes in Korean language generation’*
>
> **We thank you for your high level of expertise in language and the important points you have raised.** The proportion of morpheme combination rules in CHEF can differ depending on which morpheme analyzer is used to construct the morpheme set. As illustrated in Figures 1, 7, 8, and 9, CHEF using the mecab-ko analyzer shows a higher proportion of augmentation through regular changes.
>
> In response to your comments, **we will supplement the insufficient explanation about irregular changes on Page 4, Line 232, and add to the Limitation section about the morpheme analyzer.**
>
> The content we intend to add to the Limitation section is as follows:
>
> *'The mecab-ko analyzer, which we used for constructing the set of morphemes, can have an error rate depending on the domain and data it is applied to. The proportion of morpheme combination rules in CHEF may differ based on which morpheme analyzer is used to construct the morpheme set. Therefore, leveraging a more advanced morpheme analyzer could enable data augmentation that more accurately reflects the linguistic characteristics of the Korean language.'*
>
> We thank you for your insightful comments, which will considerably improve the paper.

---

### Meta-Review · Area_Chair_t5UH · 2023-09-18

**Recommendation:** 4

**Metareview:**

This paper proposes a data augmentation method for Korean by blending lexical and functional morphemes together to generate new forms.

The reviewers found the proposed method intuitive from a linguistic point of view and acknowledged its effectiveness on several tasks. In the initial round of reviews, the reviewers identified various weaknesses, ranging from the impact of the chosen morphological analyzer (mecab-ko) over general comprehension issues to lack of comparison with existing methods. The authors were able to engage in discussions with all three reviewers and provided additional results that, if included in the final version, would make the paper significantly stronger. All reviewers increased their scores as a result of the discussion with the authors.

---

### Decision · Program_Chairs · 2023-10-07

**Decision:**

Accept-Main

**Comment:**

This paper proposes a data augmentation method for Korean by blending lexical and functional morphemes together to generate new forms.

The reviewers found the proposed method intuitive from a linguistic point of view and acknowledged its effectiveness on several tasks. In the initial round of reviews, the reviewers identified various weaknesses, ranging from the impact of the chosen morphological analyzer (mecab-ko) over general comprehension issues to lack of comparison with existing methods. The authors were able to engage in discussions with all three reviewers and provided additional results that, if included in the final version, would make the paper significantly stronger. All reviewers increased their scores as a result of the discussion with the authors.